# 🏘️ ProcTHOR: Large-Scale Embodied AI Using Procedural Generation

**Matt Deitke**[†ψ]**, Eli VanderBilt**[†]**, Alvaro Herrasti**[†]**, Luca Weihs**[†]
**Jordi Salvador**[†]**, Kiana Ehsani**[†]**, Winson Han**[†]**, Eric Kolve**[†]
**Ali Farhadi**[ψ]**, Aniruddha Kembhavi**[†ψ]**, Roozbeh Mottaghi**[†ψ]
[†]PRIOR @ Allen Institute for AI, [ψ]University of Washington, Seattle
[procthor.allenai.org](procthor.allenai.org)

## Abstract

Massive datasets and high-capacity models have driven many recent advancements in computer vision and natural language understanding. This work presents a platform to enable similar success stories in Embodied AI. We propose PROCTHOR, a framework for procedural generation of Embodied AI environments. PROCTHOR enables us to sample arbitrarily large datasets of diverse, interactive, customizable, and performant virtual environments to train and evaluate embodied agents across navigation, interaction, and manipulation tasks. We demonstrate the power and potential of PROCTHOR via a sample of 10,000 generated houses and a simple neural model. Models trained using only RGB images on PROCTHOR, with no explicit mapping and no human task supervision produce state-of-the-art results across 6 embodied AI benchmarks for navigation, rearrangement, and arm manipulation, including the presently running Habitat 2022, AI2-THOR Rearrangement 2022, and RoboTHOR challenges. We also demonstrate strong 0-shot results on these benchmarks, via pre-training on PROCTHOR with no fine-tuning on the downstream benchmark, often beating previous state-of-the-art systems that access the downstream training data.

## 1 Introduction

Computer vision and natural language processing models have become increasingly powerful through the use of large-scale training data. Recent models such as CLIP [45], DALL-E [47], GPT-3 [7], and Flamingo [2] use massive amounts of task agnostic data to pre-train large neural architectures that perform remarkably well at downstream tasks, including in zero and few-shot settings. In comparison, the Embodied AI (E-AI) research community predominantly trains agents in simulators with far fewer scenes [46, 29, 13]. Due to the complexity of tasks and the need for long planning horizons, the best performing E-AI models continue to overfit on the limited training scenes and thus generalize poorly to unseen environments.

In recent years, E-AI simulators have become increasingly more powerful with support for physics, manipulators, object states, deformable objects, fluids, and real-sim counterparts [29, 48, 49, 19, 60], but scaling them up to tens of thousands of scenes has remained challenging. Existing E-AI environments are either designed manually [29, 19] or obtained via 3D scans of real structures [48, 46]. The former approach requires 3D artists to spend a significant amount of time designing 3D assets, arranging them in sensible configurations within large spaces, and carefully configuring the right textures and lighting in these environments. The latter involves moving specialized cameras through many real-world environments and then stitching the resulting images together to form 3D reconstructions of the scenes. These approaches are not scalable, and expanding existing scene repositories multiple orders of magnitude is not practical.

We present PROCTHOR, a framework built off of AI2-THOR [29], to procedurally generate fully-interactive, physics-enabled environments for E-AI research. Given a room specification (e.g., a

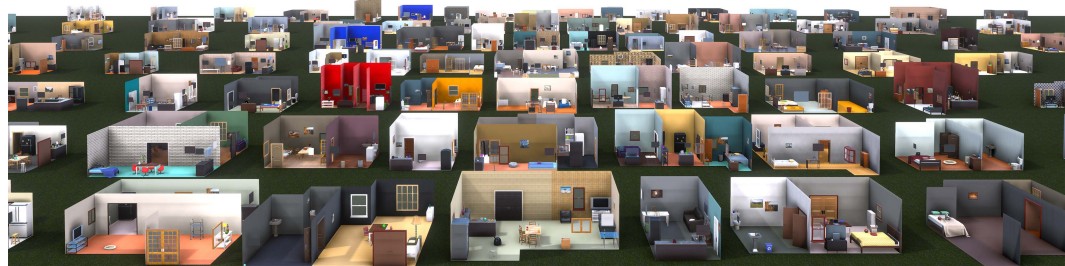

Figure 1: We propose PROCTHOR, a framework to procedurally generate a large variety of diverse, interactable, and customizable houses.

house with 3 bedrooms, 3 baths, and 1 kitchen), PROCTHOR can produce a large and diverse set of floorplans that meet these requirements (Fig. 1). A large asset library of 108 object types and 1633 fully interactable instances is used to automatically populate each floorplan, ensuring that object placements are physically plausible, natural, and realistic. One can also vary the intensity and color of lighting elements (both artificial lighting and simulated skyboxes) in each scene, to simulate variations in indoor lighting and the time of the day. Assets (such as furniture and fruit) and larger structures such as walls and doors can be assigned a variety of colors and textures, sampled from sets of plausible colors and materials for each asset category. Together, the diversity of layouts, assets, placements, and lighting leads to an arbitrarily large set of environments – allowing PROCTHOR to scale orders of magnitude beyond the number of scenes currently supported by present-day simulators. In addition, PROCTHOR supports dynamic material randomizations, whereby colors and materials of individual assets can be randomized each time an environment is loaded into memory for training. Importantly, in contrast to environments produced using 3D scans, scenes produced by PROCTHOR contain objects that both support a variety of different object states (*e.g.* open, closed, broken, *etc.*) and are fully interactive so that they can be physically manipulated by agents with robotic arms. We also present ARCHITECTHOR, a 3D artist-designed set of 10 high quality fully interactable houses, meant to be used as a test-only environment for research within household environments. In contrast to AI2-iTHOR (single rooms) and RoboTHOR (lesser visual diversity) environments, ARCHITECTHOR contains larger, diverse, and realistic houses.

We demonstrate the ease and effectiveness of PROCTHOR by sampling an environment of 10,000 houses (named PROCTHOR-10K), composed of diverse layouts ranging from small 1-room houses to larger 10-room houses. We train agents with very simple neural architectures (CNN+RNN) – *without* a depth sensor, and instead only employing RGB channels, with no explicit mapping and no human task supervision – on PROCTHOR-10K and produce state-of-the-art (SoTA) models on several navigation and interaction benchmarks. As of 10am PT on June 14th, 2022 we obtain (1) **RoboTHOR ObjectNav Challenge** [4] – 0-shot performance superior to the previous SoTA which uses RoboTHOR training scenes – with fine-tuning we obtain an 8.8 point improvement in SPL over the previous SoTA; (2) **Habitat ObjectNav Challenge 2022** [39] – top of the leaderboard results with a >3 point gain in SPL over the next best submission; (3) **1-phase Rearrangement Challenge 2022** [3] – top of the leaderboard results with Prop Fixed Strict improving from 0.19 to 0.245; (4) **AI2-iTHOR ObjectNav** – 0-shot numbers which already outperform a previous model that trains on AI2-iTHOR, with fine-tuning we achieve a success rate of 77.5%; (5) **ArmPointNav** [16] – 0-shot number that beats previous SoTA results when using RGB; and (6) **ArchitecTHOR ObjectNav** – a large success rate improvement from 18.5% to 31.4%. Finally, an ablation analysis clearly shows the advantages of scaling up from 10 to 100 to 1K and finally to 10K scenes and indicates that further improvements can be obtained by invoking PROCTHOR to produce even larger environments.

In summary, our contributions are (1) PROCTHOR, a framework that allows for the performant procedural generation of an unbounded number of diverse, fully-interactive, simulated environments, (2) ARCHITECTHOR, a new, 3D artist-designed set of houses for E-AI evaluation, and (3) SoTA results across six E-AI benchmarks covering manipulation and navigation tasks, including strong 0-shot results. PROCTHOR will be open-sourced and the code used in this work will be released.

## 2 Related Work

**Embodied AI platforms.** Various Embodied AI platforms have been developed over the past several years [29, 48, 49, 60, 19, 58]. These platforms target different design goals. AI2-THOR [29] and its variants (ManipulaTHOR [16] and RoboTHOR [13]) are built in the Unity game engine and focus on agent-object interactions, object state changes, and accurate physics simulation. Unlike

AI2-THOR, Habitat [48] provides scenes constructed from 3D scans of houses, however, objects and scenes are not interactable. A more recent version, Habitat 2.0 [51], introduces object interactions at the expense of being limited to one floorplan and synthetic scenes. iGibson [49] includes photo-realistic scenes, but with limited interactions such as pushing. iGibson 2.0 [32] extends iGibson by focusing on household tasks and object state changes in synthetic scenes and includes a virtual reality interface. ThreeDWorld [19] targets high-fidelity physics simulation such as liquid and deformable object simulation. VirtualHome [44] is designed for simulating human activities via programs. RLBench [27], RoboSuite [67] and Sapien [60] target fine-grained manipulation. The main advantage of PROCTHOR is that we can generate a diverse set of *interactive* scenes procedurally, enabling studies of data augmentation and large-scale training in the context of Embodied AI.

**Large-scale datasets.** Large-scale datasets have resulted in major breakthroughs in different domains such as image classification [15, 30], vision and language [11, 52], 3D understanding [9, 61], autonomous driving [8, 50], and robotic object manipulation [43, 40]. However, there are not many interactive large-scale datasets for Embodied AI research. PROCTHOR includes interactive houses generated procedurally. Hence, there are an arbitrarily large number of scenes in the framework. The closest works to ours are [46, 42, 33]. HM3D [46] is a recent framework that includes 1,000 scenes generated using 3D scans of real environments. PROCTHOR has a number of key distinctions: (1) unlike HM3D which includes static scenes, the scenes in PROCTHOR are interactive i.e., objects can move and change state, the lighting and texture of objects can change, and a physics engine determines the future states of the scenes; (2) it is challenging to scale up HM3D as it requires scanning a house and cleaning up the data, while we can procedurally generate more houses; (3) HM3D can be used only for navigation tasks (as there is no physics simulation and object interaction), while PROCTHOR can be used for tasks other than navigation. OpenRooms [33] is similar to HM3D in terms of the source of the data (3D scans) and dataset size. However, OpenRooms is interactive. OpenRooms is also confined to the set of scanned houses, and it takes a significant amount of time to annotate a new scene (e.g., labeling materials for one object takes 1 minute), while PROCTHOR does not suffer from these issues. Megaverse [42] is another large-scale Embodied AI platform that includes procedurally generated environments. Although it is impressive in terms of simulation speed, it includes only game-like environments with a simplified appearance. In contrast, PROCTHOR mimics real-world houses in terms of the complexity of appearance, physics, and object interactions.

**Scene synthesis.** Work on scene synthesis is typically broken down into generating floorplans [35, 36, 24, 57] and sampling object placement in rooms [17, 20, 65, 66, 12]. Our work aimed to generate diverse and semantically plausible houses using the best existing approaches or building on existing works in areas that were insufficient for our use case. Our floorplan generation process is adapted from [35, 36], which takes in a high-level specification of the rooms in a house and their connectivity constraints, and randomly generates floorplans satisfying these constraints. Our object placement is most similar to [66, 20, 65, 62, 10], where we iteratively place objects on floors, walls, and surfaces and use semantic asset groups to sample objects that co-occur (*e.g.* chairs next to tables). The modular generation process used in this work makes it easy to swap in and update any stage of our house generation pipeline with a better algorithm. In this work, we found the procedural generation approaches to be more reliable and flexible than the ones based on deep learning when adapting it to our custom object database and when generating more complex houses that were out of the distribution of static house datasets [18, 57, 34]. For a more detailed comparison, including a discussion of some of the limitations of deep learning approaches, please refer to the Appendix.

## 3 PROCTHOR

PROCTHOR is a framework to procedurally generate E-AI environments. It extends AI2-THOR and, thereby, inherits AI2-THOR's large asset library, robotic agents, and accurate physics simulation. Just as in scenes painstakingly created by designers in AI2-THOR, environments in PROCTHOR are fully interactive and support navigation, object manipulation, and multi-agent interaction.

Fig. 2 shows a high-level schematic of the procedure used by PROCTHOR to generate a scene. Given a room specification (*e.g.* house with 1 bedroom + 1 bathroom), we use multi-stage conditional sampling to, iteratively, generate a floor plan, create an external wall structure, sample lighting, and doors, then sample assets including large, small and wall objects, pick colors and textures, and determine appropriate placements for assets within the scene. We refer the reader to the appendix for details regarding our procedural generation and sampling mechanism, but highlight five key characteristics of PROCTHOR: **Diversity**, **Interactivity**, **Customizability**, **Scale**, and **Efficiency**.

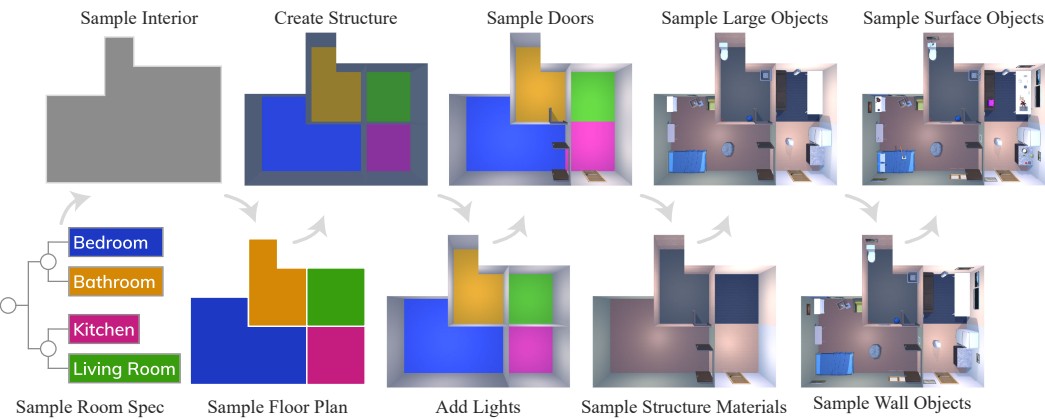

Figure 2: Procedurally generating a house using PROCTHOR.

**Diversity.** PROCTHOR enables the creation of rich and diverse environments. Mirroring the success of pre-training models with diverse data in the vision and NLP domains, we demonstrate the utility of this diversity on several E-AI tasks. Scenes in PROCTHOR exhibit diversity across several facets:

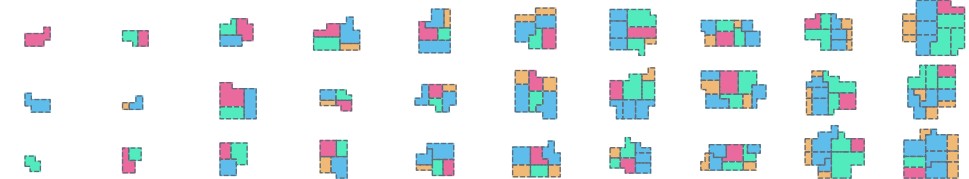

Figure 3: **Floorplan diversity.** Examples showing the diversity of the generated floorplans. Rooms in the house are colored by ■ Bedroom, ■ Bathroom, ■ Kitchen, and ■ Living Room.

*Diversity of floor plans.* Given a room specification, we first employ iterative boundary cutting to obtain an external scene layout (that can range from a simple rectangle to a complex polygon). The recursive layout generation algorithm by Lopes *et al.* [35] is then used to divide the scene into the desired rooms. Finally, we determine connectivity between rooms using a set of user-defined constraints. These procedures result in natural room layouts (e.g., bedrooms are often connected to adjoining bathrooms via a door, bathrooms more often have a single entrance, etc). As exemplified in Fig. 3, PROCTHOR generates hugely diverse floor plans using this procedure.

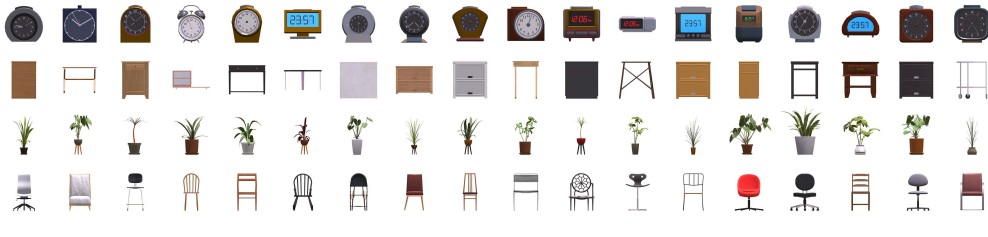

. . .

Figure 4: **Object diversity.** A subset of instances for four object categories.

*Diversity of assets.* PROCTHOR populates scenes with small and large assets from its database of 1633 household assets across 108 categories (examples in Fig. 4). While many assets are inherited from AI2-THOR, we also introduce new assets such as windows, doors, and countertops, hand-designed by 3D graphic designers. Asset instances are split into train/val/test subsets and are interactable, i.e. objects can be picked and placed within the scenes, some objects have multiple states (*e.g.* a light can be on or off) and several objects consists of parts with rigid body motions (*e.g.* door on a microwave).

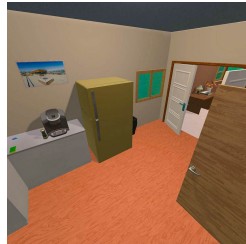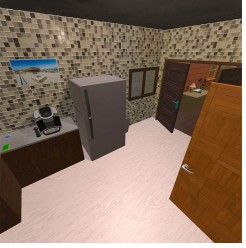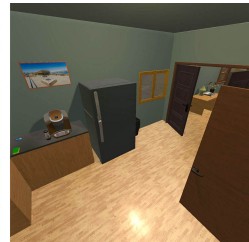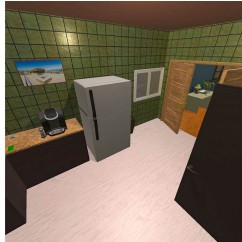

Figure 5: **Material augmentation**. Different materials for objects and structural elements.

*Diversity of materials.* Walls can have two kinds of materials – one of 40 solid (and popular) colors or one of 122 wall textures such as brick and tile. We also provide 55 floor materials. The ceiling material for the entire house is sampled from the set of wall materials. PROCTHOR also provides the ability to randomize materials of objects. Materials are only randomized within categories, which ensures objects still look and behave like the class they represent.

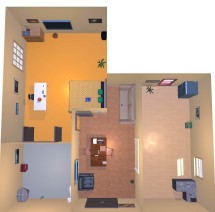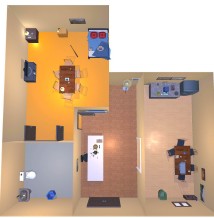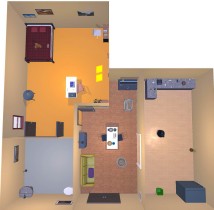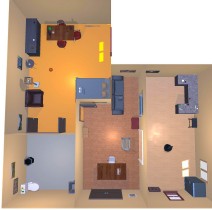

Figure 6: **Object placement.** Four examples of object placement within the same room layout.

*Diversity of object placements.* Asset categories have several soft annotations that help place them realistically within a house. These include room assignments (*e.g.* couch in a living room but not a bathroom) and location assignments (*e.g.* fridge along a wall, TV not on the floor). We also develop the notion of a Semantic Asset Group (SAG) – groups of assets that typically co-occur (*e.g.* dining table with four chairs) and thus must be sampled and placed using dependent sampling. Given a layout, individual assets and SAGs that lie on the floor are sampled and placed iteratively, ensuring that rooms continue to have adequate floor space for agents to navigate and manipulate objects. Then wall objects such as windows and paintings get placed, and finally, surface objects (ones found on top of other assets) are placed (*e.g.* cups on the kitchen counter). This sampling allows for a large and diverse set of object choices and placements within any layout. Fig. 6 shows such variations.

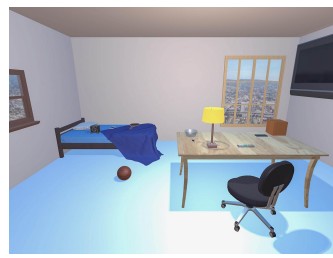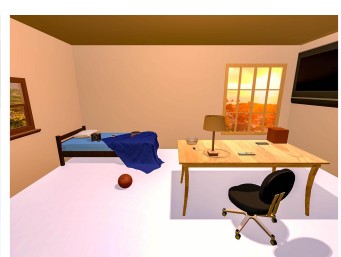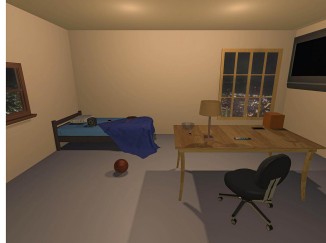

Figure 7: **Lighting variation**. Morning, dusk, and night lighting for an example scene.

*Diversity of lighting.* PROCTHOR supports a single directional light (analogous to the sun) and several point lights (analogous to lightbulbs). Varying the color, intensity, and placement of these sources allows us to simulate different artificial lighting, typically observed in houses, and also at different times of the day. Lighting has a significant effect on the rendered images as seen in Fig. 7.

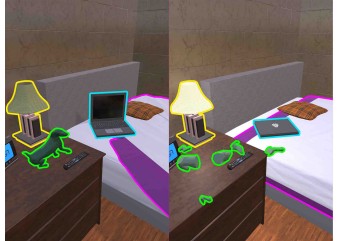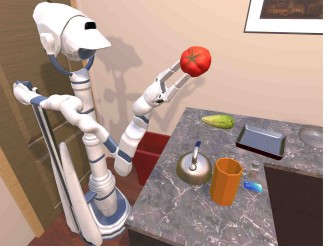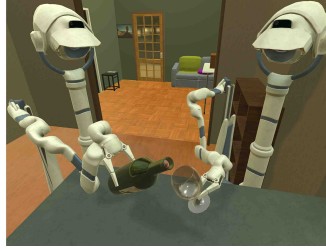

Figure 8: **Interactivity.** Object states can change (e.g., the laptop or the lamp in the left panel), and the agents can interact with objects and other agents (middle and right panels).

**Interactivity.** A key property of PROCTHOR is the ability to interact with objects to change their location or state (Fig. 8). This capability is fundamental to many Embodied AI tasks. Datasets like HM3D [46] that are created from static 3D scans do not possess this capability. PROCTHOR supports agents with arms capable of manipulating objects and interacting with each other.

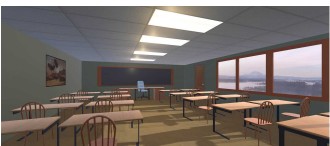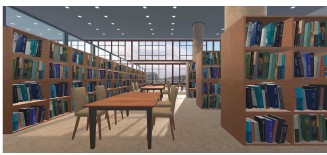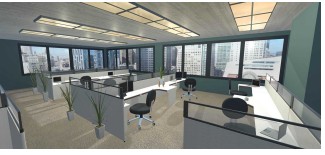

Figure 9: **Customizability.** PROCTHOR can be used to construct custom scene types such as classrooms, libraries, and offices.

**Customizability.** PROCTHOR supports many room, asset, material, and lighting specifications. With a few simple lines of specification, one can easily generate customized environments of interest. Fig. 9 shows examples of such varied scenes (classroom, library, and office).

**Scale and Efficiency.** PROCTHOR currently uses 16 different scene specifications to seed the scene generation process. These can result in over 100 billion layouts. PROCTHOR uses 18 different Semantic Asset groups and 1633 assets. These can result in roughly 20 million unique asset groups. Each of these assets can be placed in numerous locations. In addition, each house gets scaled and uses a variety of lighting. This diversity of layouts, assets, materials, placements, and lighting enables the generation of *arbitrarily large* sets of houses – either statically generated and stored as a dataset or dynamically generated at each iteration of training. Scenes are efficiently represented in a JSON specification and are loaded into AI2-THOR at runtime, making the memory overhead of storing houses incredibly efficient. Scene generation is fully automatic and fast and PROCTHOR provides high framerates for training E-AI models (see Sec. 4 for details).

## 4 PROCTHOR-10K

We demonstrate the power and potential of PROCTHOR using a sampled set of 10,000 fully interactive houses obtained by the procedural generation process described in Section 3 – which we label PROCTHOR-10K. An additional set of 1,000 validation and 1,000 testing houses are available for evaluation. Asset splits across train/val/test are detailed in the Appendix. All houses are fully navigable, allowing an agent to traverse through each room without any interaction. In terms of scale,

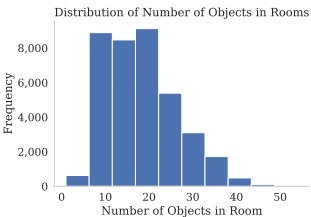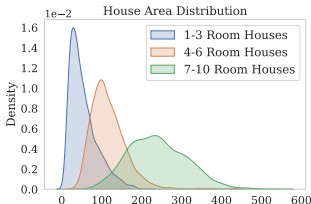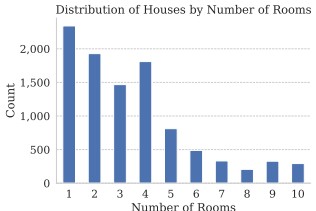

Figure 10: **PROCTHOR-10K statistics.** *Left:* distribution of the number of objects in each room; *Middle:* distribution of the area of each house, bucketed into small, medium, and large houses; *Right:* bar plot showing the distribution over the number of rooms that make up each house.

| | Navigation FPS | | Isolated Interaction FPS | | Environment Query FPS | |
|---|---|---|---|---|---|---|
| Compute | Small | Large | Small | Large | Small | Large |
| 8 GPUs | 8,599±359 | 3,208±127 | 6,488±250 | 2,861±107 | 480,205±19,684 | 433,587±18,729 |
| 1 GPU | 1,427±74 | 6,280±40 | 1,265±71 | 597±37 | 160,622±2,846 | 157,567±2,689 |
| 1 Process | 240±69 | 115±19 | 180±42 | 93±15 | 14,825±199 | 14,916±186 |

Table 1: **Rendering speed.** Benchmarking FPS for navigation (*e.g.* moving/rotating), interaction (*e.g.* pushing an object), and querying the environment for data (*e.g.* checking the dimensions of the agent). We report FPS for Small and Large houses. See Appendix for details.

PROCTHOR-10K is one of the largest sets of interactive home environments for Embodied AI – as a comparison, AI2-iTHOR [29] includes 120 scenes, RoboTHOR [13] has 89 scenes, iGibson [49] has 15 scenes, Habitat Matterport 3D [46] has 1,000 static (non-interactive) scenes, and Habitat 2.0 [51] has 105 scene layouts. Scaling beyond 10K houses is straightforward and inexpensive. This set of 10K houses was generated in 1 hour on a local workstation with 4 NVIDIA RTX A5000 GPUs. Fig. 11 shows examples of ego-centric and top-down views of houses present in PROCTHOR-10K.

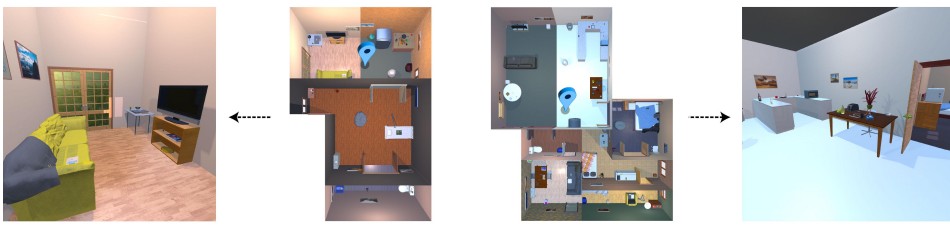

Figure 11: **Example scenes** in PROCTHOR-10K with top-down and an egocentric view.

**Scene statistics.** Houses in PROCTHOR-10K are generated using 16 different room specifications. An example room spec is: *A house with 1 bedroom connected to 1 bathroom, 1 kitchen, and 1 living room* and is visualized in Fig. 2. Houses in this dataset have as few as 1 room and as many as 10. Fig. 10 shows the distribution of areas (middle) and the number of rooms (right) of these generated houses. Our use of room specifications enables us to change the distribution of the size and complexity of houses fairly easily. PROCTHOR-10K encompasses a wider spectrum of scenes than AI2-iTHOR [29] and ROBOTHOR [13] (biased towards room-sized scenes) and Gibson [59] and HM3D [46] (biased towards large houses).

Rooms in each of these houses contain objects from 95 different categories including common household objects such as fridges, countertops, beds, toilets, and house plants, and structure objects such as doorways and windows. Fig. 10 (left) shows the distribution of the number of objects per room per house, which shows that houses in PROCTHOR-10K are well populated. They also contain objects sampled via 18 different Semantic Asset groups. Examples of Semantic asset groups (SAG) are a *Dining Table with 4 Chairs* or *Bed with 2 Pillows*. Given our large asset library and SAGs, we can create 19.3 million combinations of group instantiations.

**Rendering speed.** A crucial requirement for large-scale training is high rendering speed since the training algorithms require millions of iterations to converge. Table 1 shows these statistics. Experiments were run on a server with 8 NVIDIA Quadro RTX 8000 GPUs. For the 1 GPU experiments, we use 15 processes and for the 8 GPU experiments, we use 120 processes, evenly distributed across the GPUs. PROCTHOR provides framerates comparable to iTHOR and RoboTHOR environments in spite of having larger houses (See Appendix for details), rendering it fast enough for training large models for hundreds of millions of steps in a reasonable amount of time.

# 5  ArchitecTHOR

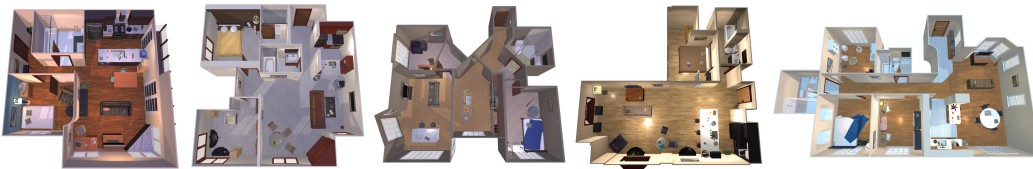

Figure 12: Top-down images of ARCHITECTHOR validation houses.

In order to test if models trained on ProcTHOR can generalize to real-world floorplans and object placements, a test set of houses was needed. Neither iTHOR (single room scenes) nor RoboTHOR (dorm-sized maze-styled scenes) contain scenes that are representative of real-world homes. Therefore, we worked with professional 3D artists to create ArchitecTHOR, which contains 10 evaluation houses (5 val, 5 test) that mimic the style of real-world homes. ArchitecTHOR val houses contain between 4-8 rooms, $121 \pm 26$ objects per house, and a typical floor size of $111 \pm 26\ m^2$. By comparison, PROCTHOR-10K houses have a much higher variance, with between 1-10 rooms, $76 \pm 48$ objects per house, and a typical floor size of $96 \pm 74\ m^2$.

# 6  Experiments

**Tasks.** We now present results for models pre-trained on PROCTHOR-10K on several navigation and manipulation benchmarks to demonstrate the benefits of large-scale training. We consider ObjectNav (navigation towards a specific object category) in PROCTHOR, ARCHITECTHOR, RoboTHOR [13], HM3D [46], and AI2-iTHOR [29]. We also consider two manipulation-based tasks: ArmPoint-Nav [16] and 1-phase Room Rearrangement [55]. In ArmPointNav, the agent moves an object using a robotic arm from a source location to a destination location specified in the 3D coordinate frame. In Room Rearrangement, the goal is to move objects or change their state to reach a target scene state.

**Models.** Our models for all tasks consist of a CNN to encode visual information and a GRU to capture temporal information. We deliberately use a simple architecture across all tasks to show the benefits of large-scale training. Our ObjectNav and Rearrangement models use the CLIP-based architectures of [28]. Our ArmPointNav model uses a simpler visual encoder with 3 convolutional layers; we found this more effective than the CLIP encoder. All models are trained with the AllenAct [56] framework, see the Appendix for training details.

**Results.** We present results in two settings: zero-shot and after fine-tuning on the training scenes provided by the downstream benchmark. Zero-shot experiments show us how well models trained on PROCTHOR generalize to new environments, whereas fine-tuning experiments tell us if representations learned from PROCTHOR can serve as a good initialization for quick tuning. For all experiments, we use only RGB images (no depth and other modalities is used).

Zero-shot is particularly challenging since other environments have different appearance statistics, layouts, and object distributions compared to PROCTHOR. ARCHITECTHOR and AI2-iTHOR [29] are high-fidelity artist-designed scenes with high-quality shadows and lighting. HM3D is constructed from 3D scans of houses which can differ quite a bit from synthetic environments. RoboTHOR [13] houses use wall panels and floors with very specific textures.

***Zero-shot transfer results.*** Models trained only on PROCTHOR and evaluated 0-shot outperform previous SoTA models on 3 benchmarks (see Table 2). These strong results suggest that models generalize to not only unseen objects and scenes, but also new appearance and layout statistics.

***Fine-tuning results.*** Further fine-tuning of the model using each benchmark's training data, achieves state-of-the-art results on all benchmarks (refer to *fine-tune* rows of Table 2). Notably, our model is ranked first on three public leaderboards as of 10am PT, June 14th 2022: Habitat 2022 ObjectNav challenge, AI2-THOR Rearrangement 2022 challenge, and RoboTHOR ObjectNav challenge. It should be noted that our model achieves these results using a very simple architecture and only RGB images. Other techniques typically use more complex architectures that include mapping or visual odometry modules and use additional perception sensors such as depth images.

| Task | Benchmark | Method | Metrics | |
|------|-----------|--------|---------|---|
| | | | Success | SPL |
| ObjectNav | RoboTHOR Challenge | EmbCLIP [28][a] | 47.0% | 0.200 |
| | | ProcTHOR 0-shot | 55.0% | 0.237 |
| | | ProcTHOR + fine-tune | **65.2%** | **0.288** |
| | | | Success | SPL |
| | | MLNLC[c] | 52.0% | 0.280 |
| ObjectNav | Habitat Challenge (2022) *HM3D-Semantics* | FusionNav (AIRI)[c] | 54.0% | 0.270 |
| | | ProcTHOR 0-shot | 9.00% | 0.055 |
| | | ProcTHOR + fine-tune | 53.0% | 0.270 |
| | | ProcTHOR + Large[d] + 0-shot | 13.2% | 0.077 |
| | | ProcTHOR + Large[d] + fine-tune | **54.4%** | **0.318** |
| | | | Success | SPL |
| ObjectNav | AI2-iTHOR | EmbCLIP [28][b] | 68.4% | 0.516 |
| | | ProcTHOR 0-shot | 75.7% | **0.644** |
| | | ProcTHOR + fine-tune | **77.5%** | 0.621 |
| | | | Success | SPL |
| ObjectNav | ARCHITECTHOR | EmbCLIP [28][b] | 18.5% | 0.118 |
| | | ProcTHOR | **31.4%** | **0.195** |
| | | | Success | % Fixed Strict |
| Rearrangement | AI2-THOR Challenge *1-phase* (2022) | EmbCLIP [28] | 7.10% | 0.190 |
| | | ProcTHOR 0-shot | 3.80% | 0.156 |
| | | ProcTHOR + fine-tune | **7.40%** | **0.245** |
| | | | Success | % PickUp SR |
| ArmPointNav | ManipulaTHOR | iTHOR-SimpleConv [16][e] | 29.2% | 73.4 |
| | | ProcTHOR 0-shot | **37.9%** | **74.8** |

Table 2: Results for models trained on ProcTHOR and evaluated 0-shot and with fine-tuning on several E-AI benchmarks. For each benchmark we also compare to the relevant baselines (previous SoTA or leaderboard submissions where applicable). [a]EmbCLIP [28] trained on ROBOTHOR, [b]EmbCLIP [28] trained on AI2-iTHOR, [c]submission on the Habitat 2022 ObjectNav leaderboard [39]. [d] For HM3D we present results when pretraining using the EmbCLIP architecture (which uses CLIP-pretrained ResNet50) as well as with a "Large" model which uses a larger CLIP backbone CNN as well as a wider RNN, see supplement for details. [e]uses the model from [16] but retrains on the complete iTHOR data with RGB inputs. ▨ 0-shot results, whereby models are pre-trained on PROCTHOR-10K and do not use any training data from the benchmark that they are evaluated on.

**Scale ablation.** To evaluate the effect of scale we train the models on 10, 100, 1k, and 10k houses. Here, we do not use any material augmentations. As shown in Table 3, the performance improves as we use more houses for training, demonstrating the benefits of large-scale data for E-AI tasks.

| | ARCHITECTHOR Test | | ROBOTHOR Test (0-Shot) | | HM3D Valid (0-Shot) | | AI2-iTHOR Test (0-Shot) | |
|---|---|---|---|---|---|---|---|---|
| # HOUSES | SPL | SR | SPL | SR | SPL | SR | SPL | SR |
| 10 Houses | 0.077 | 11.3% | 0.040 | 8.53% | 0.007 | 1.60% | 0.249 | 28.7% |
| 100 Houses | 0.102 | 18.6% | 0.076 | 20.9% | 0.050 | **10.4%** | 0.352 | 42.0% |
| 1,000 Houses | 0.122 | 17.2% | 0.157 | 33.1% | 0.027 | 4.65% | 0.456 | 53.0% |
| 10,000 Houses | **0.185** | **27.0%** | **0.210** | **44.5%** | **0.060** | 9.70% | **0.554** | **64.9%** |

Table 3: Ablation study to evaluate the effect of the number of training houses. Each model is trained to 80% success during training. Test performance increases with the number of training houses.

## 7 Conclusion

We propose PROCTHOR to procedurally generate *arbitrarily large* sets of interactive, physics-enabled houses for Embodied AI research. We pre-train simple models on 10k generated houses and show SOTA results across 6 embodied tasks with strong 0-shot results.

## Acknowledgements

We would like to thank the teams behind the open-source packages used in this project, including AI2-THOR [29], AllenAct [56], PRIOR [14], Habitat [48], 🤗 Datasets [31], NumPy [23], PyTorch [41], Pandas [38], Wandb [5], Shapely [21], Hydra [63], SciPy [53], UMAP [37], NetworkX [22], EvalAI [64], TensorFlow [1], OpenAI Gym [6], Seaborn [54], PySAT [26], and Matplotlib [25].

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
