# OpenReview forum: "🏘️ ProcTHOR: Large-Scale Embodied AI Using Procedural Generation"
_NeurIPS.cc/2022/Conference — NeurIPS 2022 Accept_

### Official Review · Reviewer_QxMz · 2022-07-08

**Rating:** 9
**Confidence:** 4
**Soundness:** 2 fair
**Presentation:** 4 excellent
**Contribution:** 3 good

**Summary:**

The work introduces ProcTHOR, a procedural indoor environment generator. Each environment generated by ProcTHOR is a plausible environment with a physics engine, which can be used for training agents to solve various indoor tasks. The authors also show that pretraining on a sample of 10k homes from this generator, either in a zero-shot setting or after fine-tuning, the trained agent outperforms top contributions on several leaderboards.

**Questions:**

- **Q.1)** I'm not sure the paper sufficiently explains what you mean by "fully interactive". I understand that objects can be modeled as rigid bodies, but do your robot agents actually do friction-based grasping, or are objects tethered to the gripper when the "grasp" action is executed with the gripper reasonably close to the object? And for shelves/fridges/things that can be opened, do they only have a binary open/closed state, or can a robot open them by an inch?
- **Q.2)** If all objects are rigid bodies, how do you assign mass, friction, and elasticity? Are these also procedural or can they be changed?
- **Q.3)** What percentage of objects have these states (open/closed, etc)?
- **Q.4)** In the appendix, you mentioned how many frames of training are required but how does that translate to wall clock time, i.e. how long did you train a policy for on ProcTHOR-10k until convergence?
- **Q.5)** It's a bit crazy to me to say that you couldn't run additional experiments to verify performance and you took the first result because they're so expensive. In RL, it's a well-known phenomenon that different seeds can lead to vastly different performance or even different implementations of the same algorithm. Could you please elaborate, or if you had time since submission, report the mean/std of your results and how many seeds were used?
- **Q.6)** Did you mention the 16 different scene specifications somewhere? Why these?
- **Q.7)** Same for the 18 different Semantic Asset groups.
- **Q.8)** You mention customizability and "a few simple lines of specifications" but what does that actually look like?
- **Q.9)** Thanks for providing the performance table in Tab.1. That's always been my main gripe with AI2THOR. But how does the process distribution work? I.e. how do you run 15 processes on each of the 8 GPUs?

**Limitations:**

~~Limitations aren't mentioned in the main body of the paper. I thought that was supposed to be included last year or 2 years ago.~~

In addition to the limitations listed in the appendix, I'd add that the lighting model is still relatively simple and might not correspond to lighting conditions in real houses but this effect may be "domain-randomized" away. Also, robot navigation is vastly simplified; real-world environments might have different friction surfaces, carpets with bumps, stairs connecting different rooms on the same floor, etc.
Also, real environments might have more decoration and trinkets lying around, as well as soft objects that can appear in many different configurations (a hoody thrown over a chair or splayed out on the sofa)

EDIT: apparently limitations can go into the appendix and the authors did just that. So this is fine.

**Strengths And Weaknesses:**

TL;DR my main concerns are (a) reproducibility/code release and (b) why this wasn't submitted to the datasets/benchmarks track.

### Strengths:
- **S.1)** The paper is really well written and structured. It was a blast to read. All the illustrations and plots are meaningful and illustrate the work.
- **S.2)** The contribution of ProcTHOR itself is great and really needed by the community.
- **S.3)** The RL results look great. The fact that with a bit of fine-tuning (and sometimes without), this outperforms many baselines is impressive.

### Weaknesses:
- **W.1)** This really should've been submitted to the dataset & benchmarks track and I don't understand why the authors didn't do that. Because now I have to evaluate this for its methodological contribution and there isn't any. To my best knowledge, the ProcTHOR generator was pieced together with various artist assets, heuristics, and a lot of human annotation. That isn't a method that translates to any other domain and the authors also make no claim that their ProcTHOR can easily be adapted to car assembly factories for example. The experiments that are shown at the end of the paper, that I positively highlighted in (S.3), aren't methodologically new. The authors even go so far as to say they deliberately used a simple agent network to highlight the benefits of the dataset. Ultimately, the decision is not on me if this fits in into the main track of the conference, so I'll refer this to the AC.
- **W.2)** Part of reviewing a paper for me is checking if the contributions claimed by the authors live up to the methods and results explained in the paper. Sadly, ArchitecTHOR (great name btw) doesn't do that. It's not even mentioned in the main body of the paper. Why is this listed as contribution if the main paper doesn't even discuss it? Sure, there are some pictures in the appendix but the main paper has to stand on its own - the appendix is for clarification and additional detail. (a) Why did you create ArchitecTHOR? (b) What did the designers focus on in designing these spaces? (c) What wasn't there yet in AI2THOR that needed to be added here? (d) What are the statistics of the spaces in terms of floor size, rooms, number of objects? (e) How does that compare to envs generated by ProcTHOR? (f) When should I use one over the other for training or is A-THOR only for evaluation? If these questions are answered in the main body of the paper, I'm happy to recognize ArchitecTHOR as dataset contribution and increase my score.
- **W.3)** The authors make the vague promise that at some point, code will be released publicly. This is a dataset paper and I have no way to reproduce the hundreds if not thousands of person-hours that went into this. If this is not reproducible and not publicly available, it's of no use to the scientific community. If this was a paper introducing a new method, this would be different because, from the mathematical description of the method, I'd be able to at least reproduce some of it. Since this is not the case, I have to make a deduction in my rating. I have to insist that you release the dataset (generator) if you want us to accept a dataset paper.
- **W.4)** In (S.1) I praised the writing and structure and I stand by that, but this was made at the sacrifice of some essential details. Sadly most of the information on HOW ProcTHOR was made was relegated to the appendix. I understand that this is due to space constraints but it kind of diminishes the value of the main paper with respect to the appendix.
- **W.5)** Validity of claims: In the related works section, you mention HM3D, OpenRooms, and Megaverse and your main criticism of these is that they're either too small or too game-like but you don't verify that claim as far as I can tell. A way to verify this would be to use these environments for pretraining, instead of ProcTHOR-10k, and check the 0-shot and fine-tuning performance, at least in object-nav tasks (because yes, some benchmarks require interaction, and that's not offered in these but that's beside the point). This way, you can verify that (a) the size/diversity of ProcTHOR-10k is necessary and (b) the visual fidelity of ProcTHOR is necessary for this demonstrated transfer performance.

---

> ### Author Response · Authors · 2022-08-02
> **Reviewer QxMz Response - Part 1**
>
> Thank you for the insightful and valuable feedback. We appreciate the
> positive comments that the paper is well-written (“it was a blast to
> read”), ProcTHOR is a great contribution to the community, and the
> results are impressive.
>
> > TL;DR my main concerns are (a) reproducibility/code release and
> > (b) why this wasn't submitted to the datasets/benchmarks track.
>
> We first address the two concerns listed in the TL;DR and then address
> all other concerns below.
>
> > W.1) Why this wasn't submitted to the datasets/benchmarks track.
>
> We have addressed this concern in the Overall Response seen above.
> Please refer to that response.
>
> > W.3) Dataset (generator) release.
>
> We appreciate your concern with releasing the dataset generator. We
> mention in L78 of the original submission that “ProcTHOR will be
> open-sourced and the code used in this work will be released” and
> we fully stand by that. In fact, all of our code is prepared for
> release, linked anonymously here:
> - ProcTHOR House Generation Code:
> [https://anonymous.4open.science/r/procthor-CF0D/](https://anonymous.4open.science/r/procthor-CF0D/).
> - The ProcTHOR-10K dataset, consisting of the 10K houses used to
> train the agents in this paper, is available here:
> [https://anonymous.4open.science/r/procthor-10k-27FB/](https://anonymous.4open.science/r/procthor-10k-27FB/)
> - The code to train the agents is available here: [https://anonymous.4open.science/r/procthor-training-6FDD](https://anonymous.4open.science/r/procthor-training-6FDD)
>
> After the double-blind review period has concluded, we will share the
> link to the open-sourced code-base (licensed under the Apache 2.0
> license, making the assets and scenes broadly available for both
> commercial and non-commercial work). We commit to withdrawing the
> paper if the code is not available by September 14th.
>
> > The ProcTHOR generator does not translate to any other domain.
>
> The human annotation that was required to create the scene generator
> for houses was roughly several hours once the infrastructure was in
> place. This included creating semantic asset groups, labeling where
> object types can be placed, and creating room specs. Our motivation
> for procedurally generating houses was to create a strong pre-training
> dataset suitable for well studied downstream tasks in Embodied AI,
> which tend to presently focus on household environments.
>
> However, adapting the scene generator to environments beyond houses
> is fairly minimal and reasonably fast. For instance, to generate
> classrooms, one could use the same process, just defining new room
> specs, potentially adding and labeling new objects, and creating
> semantic asset groups for co-pairings such as  like chairs attached
> to desks.

---

> > ### Author Response · Authors · 2022-08-02
> > **Reviewer QxMz Response - Part 2**
> >
> > > W.2) If my questions on ArchitecTHOR are answered in the main body
> > > of the paper, I'm happy to recognize ArchitecTHOR as dataset
> > > contribution and increase my score.
> >
> > We appreciate you pointing out that as a key contribution, ArchitecTHOR
> > should be discussed in the main body of the paper. This is a fair point.
> > Given the large number of contributions and visuals, we chose to highlight
> > the key findings in the main body of the paper. This included an overview
> > of ProcTHOR, its key features and the experimental results obtained by
> > models employing pre-training via ProcTHOR. Contributions such as
> > ArchitecTHOR, technical details and qualitative results were just as
> > important, but, due to the space constraints, were presented in the
> > supplementary material. Given your feedback, we have revised our main
> > paper to include ArchitecTHOR and we briefly discuss this section in
> > the context of your questions below.
> >
> > **(a) Why did you create ArchitecTHOR?** Since ProcTHOR is procedurally
> > generated, we needed a test set of houses that were drawn from a
> > real-world distribution to test if models trained on ProcTHOR merely
> > memorized biases from the procedural generation, or if they were
> > capable of generalizing to real-world floorplans and object placements.
> >
> > **(b) What did the designers focus on in designing these spaces?**
> > Designers were tasked with designing houses that mimicked real-world
> > homes and were encouraged to pick and place assets that are typically
> > observed within such homes. They did not have access to the procedurally
> > generated scenes when they designed ArchitecTHOR.
> >
> > **(c) What wasn't there yet in AI2THOR that needed to be added here?**
> > AI2-THOR includes 2 interactive scene datasets: iTHOR and RoboTHOR.
> > iTHOR contains single-room-sized scenes whereas RoboTHOR includes
> > dorm-sized maze-styled scenes that are not representative of
> > real-world-sized and styled homes. Neither of these represented
> > real-world houses that typically contain many rooms, which is why
> > we chose to hire professional 3D artists to create ArchitecTHOR.
> >
> > **(d) What are the statistics of the spaces in terms of floor size,
> > rooms, number of objects?** ArchitecTHOR validation houses contain
> > between 4-8 rooms, 121.4 ± 26.1 objects per house, and a typical
> > floor size of 111.1 ± 26.4 m².
> >
> > **(e) How does that compare to envs generated by ProcTHOR?** By comparison,
> > ProcTHOR-10K houses have a much higher variance, with between 1-10 rooms,
> > 75.7 ± 48 objects per house, and a typical floor size of 95.6 ± 74.2 m².
> >
> > **(f) When should I use one over the other for training or is A-THOR
> > only for evaluation?** ArchitecTHOR is meant to be used only for
> > evaluation given the few number of scenes. Using these for training
> > will likely result in overfitting to those 10 houses.
> >
> > > W.4) Sadly most of the information on HOW ProcTHOR was made was
> > > relegated to the appendix.
> >
> > As mentioned above in response to your weakness W.2, our paper
> > contains several contributions, details and visuals, all of which were
> > deemed important to present to the reviewers and readers. However,
> > due to the space constraints set by NeurIPS, we had to be selective
> > about what went into the main paper and what was added to the appendix.
> >
> > We posited that the majority of readers of our paper would be interested
> > in learning more about the features of ProcTHOR to determine if it
> > would be applicable to their research and a fewer but important
> > number of readers would be interested in the technical details of
> > how one creates such environments. Similarly, we posited that the
> > majority of our readers would be interested in seeing our strong
> > Embodied AI results across tasks and simulators, and a fewer but
> > important number would be interested in finer details such as the
> > noise model employed during motion.
> >
> > In addition, technical details for large projects like ProcTHOR,
> > while valuable in text, can only be completely detailed via a
> > code release. As outlined above, we assure you that the entire
> > codebase for ProcTHOR and our experiments will be made open source.
> >
> > These were the reasons why technical details for ProcTHOR were
> > added to the appendix.

---

> > > ### Comment · Reviewer_QxMz · 2022-08-07
> > > **I don't know why every response in a thread on Openreview has to have its own title.**
> > >
> > > I saw that you added the new section on ArchitecTHOR into the main body of the paper and I appreciate that. Will update the score accordingly.
> > >
> > > Regarding "...the majority of readers of our paper would be interested in learning more about the features of ProcTHOR..." and "...fewer [...] readers would be interested in the technical details..." - aren't you now making the case here that this is more of a dataset paper and it's more about the features of the dataset and less about the technical contribution?

---

> > ### Author Response · Authors · 2022-08-02
> > **Reviewer QxMz Response - Part 3**
> >
> > > W.5) Validity of claims -- other simulators too small to pre-train.
> >
> > Thank you for this helpful suggestion. In the table below we report
> > the zero-shot performance of ProcTHOR, iTHOR, and HM3D pretrained
> > models on the ArchitecTHOR validation set. For the HM3D model we
> > also report performance after fine-tuning on iTHOR. Here we use a
> > model trained for 195M steps in HM3D, which attains a success rate
> > of 53% success rate and an SPL of 0.3045 on a 200 episode subset
> > of the validation set (this checkpoint was chosen as it maximized
> > the SPL). We notice that leveraging HM3D’s additional data for
> > pre-training improves performance over training without it, but
> > is still significantly outperformed by just training on ProcTHOR.
> >
> > | Model | ArchitecTHOR Val Success | ArchitecTHOR Val SPL |
> > |-------|-------|-------|
> > |**ProcTHOR 0-Shot** | **63.1%** | **0.469** |
> > |HM3D 0-Shot | 8.89% | 0.041 |
> > | HM3D Fine-Tuned on iTHOR | 45.9%| 0.3503 |
> > | iTHOR 0-Shot | 36.3% | 0.262|
> >
> > For OpenRooms, the 3D models of the scenes are not publicly
> > available to use. Therefore we unfortunately cannot use their
> > scenes for any robotics experiments.
> >
> > Megaverse only includes toy environments that consist only o
> > f primitive objects, such as cubes and cylinders
> > ([paper](https://arxiv.org/pdf/2107.08170.pdf)). Thus,
> > it is most unlikely that ObjectNav agents pre-trained on
> > Megaverse will be effective in household-based tasks.
> >
> > The above results indicate that the size and diversity of
> > ProcTHOR-10k hugely benefits transfer performance.
> >
> > > Q.1) what do you mean by "fully interactive".
> >
> > ProcTHOR inherits all its interactive functionality from AI2-THOR.
> > It currently supports manipulation that abstracts away friction-based
> > grasping. Objects are attached to the gripper when the gripper is
> > sufficiently close and the grasp action is called (see the ManipulaTHOR
> > paper for more details on that agent). The open/close state is not
> > binary, as openable objects can be opened fractionally by any amount.
> > There is also support for the ManipulaTHOR agent opening doors
> > inch-by-inch (for an example, see:
> > [https://procthor-rebuttal.netlify.app/arm-open-close.mp4](https://procthor-rebuttal.netlify.app/arm-open-close.mp4)).
> >
> >
> > > Q.2) If all objects are rigid bodies, how do you assign mass, friction, and elasticity? Are these also procedural
> > > or can they be changed?
> >
> > For both the assets used in AI2-THOR’s asset library and our custom-built
> > assets, such properties are manually specified on a per-asset basis,
> > which is estimated based on the values of similar real-world objects.
> > However, the simulator also supports changing these values to arbitrary
> > numbers at runtime. This functionality can support new research
> > directions (e.g. requiring agents to estimate the mass of objects
> > by pushing them).
> >
> > > Q.3) What percentage of objects have these states (open/closed, etc)?
> >
> > Among the 1,633 objects currently in our object database:
> > - Pickupable: 678 / 1633 ~ 41.5%
> > - Openable: 186 / 1633 ~ 11.4%
> > - Moveable: 588 / 1633 ~ 36% - note that objects like chairs may be
> >   moved but not picked up by any of AI2-THOR’s current agents
> > - Breakable: 217 / 1633 ~ 13.3%
> > - Transparent: 31 / 1633 ~ 1.9%
> > - Switched on or off: 281 / 1633 ~ 17.2%
> > - Cookable: 30 / 1633 ~ 1.8%
> > - Heat surfaces (e.g., microwaves that can cook objects): 90 / 1633 ~ 5.5%
> > - Cold surfaces (e.g., fridges that can freeze objects): 30 / 1633 ~ 1.8%
> >
> > > Q.4) What is the wall-clock time for ProcTHOR training?
> >
> > Section F of the appendix contains details regarding the wall clock times
> > for each of the experiments. To summarize:
> > - L532: ProcTHOR ObjectNav pre-training takes 5 days for 423 million steps.
> > - L564: RoboTHOR ObjectNav fine-tuning takes 7 hours for 29 million steps.
> > - L571: HM3D-Semantic ObjectNav fine-tuning takes 43 houses for 220 million steps.
> > - L578: AI2-iTHOR ObjectNav fine-tuning takes 1.5 hours for 2 million steps.
> > - L593: ProcTHOR ArmPointNav takes 3 days for 100M steps.
> > - L611: ProcTHOR Rearrangement pre-training takes 4 days for 182 million steps.
> > - L617: AI2-iTHOR Rearrangement fine-tuning takes 16 hours for 9 million steps.
> >
> > Note that the line numbers correspond to those in the originally submitted
> > supplementary materials.
> >
> > > Q.5) Random seeds
> >
> > We reran ProcTHOR ObjectNav pre-training with 5 different random seeds
> > for 100M steps and found that the variance across seeds is quite small.
> > This measurement was performed for our 0-shot results on a set of
> > 1000 ObjectNav tasks divided evenly between unseen ProcTHOR val
> > homes, ArchitecTHOR val, iTHOR val, and RoboTHOR val.
> >
> > We obtained:
> > - Train success: 0.6787 mean, 0.0289 std
> > - Val success: 0.453 mean, 0.012 std
> >
> > (Here, Train numbers refer to ProcTHOR train and Val refers to the 1000
> > task set detailed above.)
> >
> > The train and val curves across different random seeds also follow
> > each other closely. Here is an anonymous link to images of them:
> > [https://procthor-rebuttal.netlify.app/random-seeds](https://procthor-rebuttal.netlify.app/random-seeds).

---

> > > ### Comment · Reviewer_QxMz · 2022-08-07
> > > **Response**
> > >
> > > Great. I'd recommend adding the response to W.5 into the paper to make it slightly stronger.
> > > And the additional experiments and response to Q.5 are appreciated!

---

> > > > ### Author Response · Authors · 2022-08-09
> > > > **Reviewer QxMz Rebuttal Response**
> > > >
> > > > We thank the reviewer for their thoughtful feedback and support of the work.
> > > >
> > > > > I'd recommend adding the response to W.5 into the paper to make it slightly stronger.
> > > >
> > > > Thank you for the suggestion. We agree and have added the results to Section H of the Appendix.

---

> > ### Author Response · Authors · 2022-08-02
> > **Reviewer QxMz Response - Part 4**
> >
> > > Q.6) Did you mention the 16 different scene specifications somewhere?
> > > Why these?
> >
> > It’s in the attached code [here](https://anonymous.4open.science/r/procthor-CF0D/procthor/generation/room_specs.py).
> > We empirically found these room specs to be representative of the vast majority of single-floor houses
> > and supported generating highly diverse and plausible houses. Please also see our answer
> > to Reviewer 2 (W2/Q2) regarding what these specifications correspond to.
> >
> > > Q.7) Same for the 18 different Semantic Asset groups.
> >
> > We manually created semantic asset groups based on plausible
> > relationships between objects (such as chairs around tables,
> > televisions on stands, and night stands next to beds). The
> > full list is defined in the `asset_groups` folder [here](https://anonymous.4open.science/r/procthor-CF0D/procthor/databases/__init__.py).
> >
> > > Q.8) You mention customizability and "a few simple lines of
> > > specifications" but what does that actually look like?
> >
> > Here is a full example of what a room spec for a kitchen and
> > living room looks like:
> >
> > ```python
> > RoomSpec(
> >     room_spec_id="kitchen-living-room",
> >     sampling_weight=2,
> >     spec=[
> >         LeafRoom(room_id=2, ratio=1, room_type="Kitchen"),
> >         LeafRoom(room_id=3, ratio=1, room_type="LivingRoom")
> >     ]
> > )
> > ```
> >
> > and for a house with a bathroom, bedroom, kitchen, and living room:
> >
> > ```python
> > RoomSpec(
> >     room_spec_id="4-room",
> >     sampling_weight=5,
> >     spec=[
> >         MetaRoom(
> >             ratio=2,
> >             children=[
> >                 LeafRoom(room_id=4, ratio=2, room_type="Bedroom"),
> >                 LeafRoom(room_id=5, ratio=1, room_type="Bathroom")
> >             ],
> >         ),
> >         MetaRoom(
> >             ratio=2,
> >             children=[
> >                 LeafRoom(room_id=6, ratio=3, room_type="Kitchen"),
> >                 LeafRoom(room_id=7, ratio=2, room_type="LivingRoom")
> >             ]
> >         )
> >     ]
> > )
> > ```
> >
> > Terms used:
> > - room_spec_id: uniquely identifies the room spec amongst a set of room
> > specs.
> > - sampling_weight: specifies how frequently a given room spec should
> > be sampled, relative to others in a set of room specs.
> > - room_id: makes it easy to debug the mapping between rooms in the
> > room spec and those in the final house.
> > - ratio: discussed in the floorplan generation section of the appendix.
> > - room_type: the type of room, such as kitchen or bathroom.
> > - MetaRoom: defines a room “subtree”, which contains two or more
> > child rooms.
> > - LeafRoom: a room that does not have any child rooms.
> >
> > > Q.9) Thanks for providing the performance table in Tab.1. That's
> > > always been my main gripe with AI2THOR. But how does the process
> > > distribution work? I.e. how do you run 15 processes on each of
> > > the 8 GPUs?
> >
> > ProcTHOR takes up 400 MB of space each time it is initialized to
> > create a new Unity window. The 400 MB of space can be allocated
> > to any 8 of the GPUs. For our experiments, we create 15 x 8 = 120
> > instances of ProcTHOR, evenly distributed across the 8 GPUs. On
> > each GPU, the 15 instances take up about 400 MB x 15 = 6 GB of space.
> > We then use Python’s multiprocessing module to execute actions on
> > each of the 120 instances in parallel, which is similar to how
> > agents are trained in parallel with AllenAct.
> >
> > ### Limitations
> >
> > > Limitations aren't mentioned in the main body of the paper. I
> > > thought that was supposed to be included last year or 2 years ago.
> >
> > To the best of our knowledge, limitations do not have to appear
> > in the main body of the paper. The
> > [2022 style guide](https://media.neurips.cc/Conferences/NeurIPS2022/Styles/neurips_2022.pdf)
> > only mentions limitations in L171 and does not make reference to
> > it being in the main paper.
> >
> > >I'd add that the lighting model is still relatively simple and might not correspond to lighting conditions in real houses but this effect may be "domain-randomized" away.
> >
> > >Also, robot navigation is vastly simplified; real-world environments might have different friction surfaces, carpets with bumps, stairs connecting different rooms on the same floor, etc.
> >
> > >Also, real environments might have more decoration and trinkets lying around, as well as soft objects that can appear in many different configurations (a hoody thrown over a chair or splayed out on the sofa).
> >
> > There is a long way to go both in simulation and agent modeling before
> > we can have faithful virtual worlds that can be considered
> > interchangeable with the real-world, and agents that can properly
> > react to all that variability. However, we believe that ProcTHOR
> > will enable progress in this direction.

---

> > > ### Comment · Reviewer_QxMz · 2022-08-07
> > > **Re: Re: Re: Re: Response**
> > >
> > > Regarding limitations: alright fair enough. My mistake - I'll adjust that in the review.
> > >
> > > Regarding answers: okay, these answers are great and I appreciate the Q.8 code example.

---

> > ### Comment · Reviewer_QxMz · 2022-08-07
> > **Appreciate Code**
> >
> > I appreciate the commitment of the authors and the promise in combination with the anonymous code convinces me that code will be released.

---

### Official Review · Reviewer_AqAB · 2022-07-12

**Rating:** 8
**Confidence:** 4
**Soundness:** 4 excellent
**Presentation:** 4 excellent
**Contribution:** 3 good

**Summary:**

The paper presents ProcThor, a framework to sample virtual and interactive 3D environments from an underlying distribution of room and object layouts. In the current work, 10000 3D environments of varying sizes, # rooms, and object distributions are sampled and enable simulation for object search and manipulation tasks. The experiments demonstrate that training policies on the synthetically generated environments and the finetuning it on other datasets like RoboThor, AI2Thor, and Habitat-Matterport3D lead to state-of-the-art performances. Furthermore, the ablation experiments reveal that the transfer task performance continues to improve as more and more virtual 3D environments are sampled for training.

**Questions:**

- Is the scene generation process novel? Could the authors do a detailed comparison of different steps to existing literature? This is essential for understanding ProcTHOR and improving it in the future.
- How are room specs obtained (and are they realistic)? Does having only 16 specs limit the diversity?
- How do rendering speeds compare to other frameworks like AI2Thor, iGibson, Gibson, Habitat, Habitat-2.0, etc?
- What is being transferred when the visual appearance is significantly different (like HM3D-Semantic ObjectNav)?
- Do the scaling ablations hold true when models are finetuned? Does the lack of consistent scaling for HM3D-Semantic ObjectNav reflect poorly on the ability to use ProcThor to benefit real-world robotics?


**Limitations:**

The authors have adequately addressed this.

**Strengths And Weaknesses:**

# ----------------------- Post-rebuttal update --------------------
The authors' responses during the rebuttal addressed majority of my concerns and I'm retaining my rating at 8. I request that the authors update the paper with the reviewer feedback where appropriate.

I'd also like to see the HM3D-objectnav scaling experiments under fine-tuning settings in the final version of the paper (or even in supp). I feel like that would partially answer the question of whether synthetically generating training environments can enable meaningful performance-scaling in real-world like scenes (with very different visual characteristics).


# ----------------------- Pre-rebuttal review --------------------
# Strengths
- The paper is well-written and easy to understand. The supplementary gives detailed information about generating scenes and helps clarify any ambiguity.
- The framework developed is very extensive and is a massive feat of engineering (as detailed in supplementary). The authors have shared the code for their experiments and promised to open source it. I expect this to be very impactful for the embodied AI literature and beyond.
- L208 - The efficiency of generating scenes is also impressive. 10k scenes were generated in 1 hour with 4 GPUs. The scenes are also efficient to store in a small JSON file per scene.
- The experiments are well designed to cover 3 tasks from multiple online leaderboards. Both the zero-shot and finetuning experiments are impressive, especially considering that the agent only relies on RGB sensing (and not depth or panoramic sensing like some prior work).

# Weaknesses
- The novelty of the scene generation process is a bit unclear. My impression is that regardless of the novelty of each step of the pipeline, this framework on its own is a massive feat of engineering. Nevertheless, it would be useful to do a careful related works comparison to different steps of the pipeline. For example, the Semantic Asset Groups are similar to hyper-relations from [1].
- In supplementary Sec. B2, the authors state the use of only 16 room specs for generating the 10000 scenes. This raises a question of diversity for generation. It's also not clear how room specs were obtained in the first place.
- Table 1 - the rendering speeds are good, but there is no comparison to existing simulation platforms like iGibson, Gibson, Habitat, Habitat-2.0, etc. I feel it is important to benchmark these under a consistent hardware setup. It is okay even if ProcThor is slower than any of these individual frameworks since none of them support procedural generation of interactive scenes (and on such a large scale).
- Table 2 - in the transfer results to HM3D-Semantic ObjectNav, the zero-shot performance is very low, but the fine-tuned performance is state-of-the-art. I'm not sure I understand what is being transferred here (visual representations? Navigation ability? Object-based scene priors? Ability to stop?)
- Table 3 - the ablation study is good, but it leaves two questions unanswered. The results are all zero-shot, so it is not clear if the trends hold true when models are finetuned. Also, the HM3D zero-shot performance isn't consistently increasing. This raises the question of whether this large-scale procedural generation will actually help for real-world robotics.

[1] Zhang, Shao-Kui, Wei-Yu Xie, and Song-Hai Zhang. "Geometry-based layout generation with hyper-relations among objects." Graphical Models 116 (2021): 101104.

---

> ### Author Response · Authors · 2022-08-02
> **Reviewer AqAB Response - Part 1**
>
> Thank you for the insightful and valuable feedback. We appreciate the
> positive comments that the work is very impactful for the embodied
> AI literature and beyond, the efficiency of generating scenes is
> impressive, both the zero-shot and fine-tuning experiments are
> impressive, and the paper is well-written.
>
>
> > W1 and Q1) Is the scene generation process novel? Could the authors do
> > a detailed comparison of different steps to existing literature?
> > This is essential for understanding ProcTHOR and improving it in the
> > future.
>
>
> Thank you for the suggestion and the paper reference. We agree that a
> more detailed comparison would be useful. Therefore, we have updated
> the scene synthesis section of the related works in Section 2 and
> Appendix B.12 now includes a detailed comparison of the different
> steps of our scene generation process to the literature.
>
> To summarize, work on scene synthesis is typically broken down into
> generating floorplans and sampling object placement in rooms. Our
> work aimed to generate diverse and semantically plausible houses
> using the best existing approaches or building on existing works
> in areas that were insufficient for our use case. Our floorplan
> generation process is adapted from [1,2], which takes in a high-level
> specification of the rooms in a house and their connectivity
> constraints, and randomly generates floorplans satisfying these
> constraints. Our object placement is most similar to [3, 4, 5, 6, 7],
> where we iteratively place objects on floors, walls, and surfaces and
> use semantic asset groups to sample objects that co-occur (e.g.,
> chairs next to tables). The modular generation process used in this
> work makes it easy to swap in and update any stage of our house
> generation pipeline with a better algorithm. In this work, we found
> the procedural generation approaches to be more reliable and flexible
> than the ones based on deep learning when adapting it to our custom
> object database and when generating more complex houses that were out
> of the distribution of static house datasets [8, 9, 10]. For a more
> detailed comparison, including a discussion of some of the limitations
> of deep learning approaches, please refer to the Appendix B.12.
>
> [1] Lopes, R., Tutenel, T., Smelik, R. M., De Kraker, K. J., & Bidarra, R. (2010, November). A constrained growth method for procedural floor plan generation. In Proc. 11th Int. Conf. Intell. Games Simul (pp. 13-20). Citeseer.
>
> [2] Marson, F., & Musse, S. R. (2010). Automatic real-time generation of floor plans based on squarified treemaps algorithm. International Journal of Computer Games Technology, 2010.
>
> [3] Zhang, S. K., Xie, W. Y., & Zhang, S. H. (2021). Geometry-based layout generation with hyper-relations among objects. Graphical Models, 116, 101104.
>
> [4] Germer, T., & Schwarz, M. (2009, December). Procedural Arrangement of Furniture for Real‐Time Walkthroughs. In Computer Graphics Forum (Vol. 28, No. 8, pp. 2068-2078). Oxford, UK: Blackwell Publishing Ltd.
>
> [5] Yu, L. F., Yeung, S. K., Tang, C. K., Terzopoulos, D., Chan, T. F., & Osher, S. J. (2011). Make it home: automatic optimization of furniture arrangement. ACM Transactions on Graphics (TOG)-Proceedings of ACM SIGGRAPH 2011, v. 30,(4), July 2011, article no. 86, 30(4).
>
> [6] Xu, K., Chen, K., Fu, H., Sun, W. L., & Hu, S. M. (2013). Sketch2Scene: Sketch-based co-retrieval and co-placement of 3D models. ACM Transactions on Graphics (TOG), 32(4), 1-15.
>
> [7] Chang, A., Savva, M., & Manning, C. D. (2014, October). Learning spatial knowledge for text to 3D scene generation. In Proceedings of the 2014 conference on empirical methods in natural language processing (EMNLP) (pp. 2028-2038).
>
> [8] Fu, H., Cai, B., Gao, L., Zhang, L. X., Wang, J., Li, C., ... & Zhang, H. (2021). 3d-front: 3d furnished rooms with layouts and semantics. In Proceedings of the IEEE/CVF International Conference on Computer Vision (pp. 10933-10942).
>
> [9] Wu, W., Fu, X. M., Tang, R., Wang, Y., Qi, Y. H., & Liu, L. (2019). Data-driven interior plan generation for residential buildings. ACM Transactions on Graphics (TOG), 38(6), 1-12.
>
> [10] LIFULL Co., Ltd. (2015): LIFULL HOME'S Dataset. Informatics Research Data Repository, National Institute of Informatics. (dataset). https://doi.org/10.32130/idr.6.0

---

> > ### Author Response · Authors · 2022-08-02
> > **Reviewer AqAB Response - Part 2**
> >
> > > W2 and Q2) Does having only 16 specs limit the diversity?
> >
> > Room specs are quite simple and abstract, a single room spec outlines
> > the rooms present in a house along with some connectivity constraints.
> > For example, a single room spec might be a house with 3 beds, 2 baths,
> > a kitchen, and a living room. As these specs are so generic, they can
> > generate an unbounded set of houses with unique floorplans and object
> > placements. Hence, while using 16 specs does impose some constraints
> > on the types of houses that can be generated (e.g. we did not have a
> > "house" that is just two connected bathrooms), the amount of diversity
> > is still extremely high.
> >
> > If downstream tasks and environments contain houses unsupported by the
> > present 16 specs, practitioners can easily add new specs manually
> > and generate large numbers of diverse houses pertaining to those new specs.

---

> > > ### Comment · Reviewer_AqAB · 2022-08-05
> > > **Rebuttal response from reviewer AqAB**
> > >
> > > I thank the authors for their detailed responses and experiments to address the reviewer concerns. Most of my concerns are addressed and I'm inclined to retain my rating. Please see detailed responses below.
> > >
> > > ### Related work comparison for scene generation
> > > * This addresses my concern. Thanks.
> > >
> > > ### What is being transferred for HM3D ObjectNav?
> > > * In summary, the authors suggest that navigational behaviors are transferred despite the visual appearance change. The train/val curves appear to indicate this.
> > > * For a more thorough analysis, I would suggest that the authors perform interpretability studies like: https://openaccess.thecvf.com/content/CVPR2022/html/Dwivedi_What_Do_Navigation_Agents_Learn_About_Their_Environment_CVPR_2022_paper.html
> > >
> > > ### Do scaling ablations hold true for fine-tuning?
> > > * It is encouraging to see benefits of scaling for RoboTHOR fine-tuning. But I would have preferred to see results on HM3D fine-tuning instead since that's where the 0-shot performance is not very indicative. It would also be more indicative of benefit of ProcThor for real-world robotics.
> > > * If there is sufficient time, could the authors try this? I think this is important, particularly since the authors state that the 0-shot performance scaling for HM3D is unreliable.
> > >
> > > ### Speed comparisons with other frameworks
> > > * Thanks for these comparisons. It is good to see the navigational simulation speed is somewhat comparable with that of Habitat HM3D.
> > > * For the final version of the paper, it would be nice to have the navigation + interaction simulation speed comparison with Habitat-2.0 / iGibson.
> > >
> > > ### Do RoomSpecs limit diversity?
> > > * Understood. Thanks for the clarification.
> > >
> > > As a final note, I'm tending towards retaining my high-score despite the other reviews since my take on the paper is a bit different. I agree that the techniques used in ProcTHOR may not be completely novel. Also, given the scope of the effort taken, the details in the main paper may not be very satisfying with a 9-page limit. However, I feel that the biggest contributions from ProcTHOR are:
> > > * the engineering effort of putting together the best methods from this vast scene generation literature and coming up with an effective tool
> > > * demonstrating convincingly that it is effective for EAI on several tasks
> > >
> > > Given the amount of effort spent on the ProcTHOR pipeline, it would have been tempting to go light on the experiments to demonstrate its utility. So I appreciate the experimental thoroughness despite the weaknesses cited so far (many of which have been eventually addressed anyway).

---

> > > > ### Comment · Reviewer_QxMz · 2022-08-07
> > > > **Re "retaining my high-score despite the other reviews"**
> > > >
> > > > Dear R#AqAB,
> > > >
> > > > > the engineering effort of putting together the best methods from this vast scene generation literature and coming up with an effective tool
> > > >
> > > > I agree that this was a massive undertaking and should be recognized as such. But I feel like for "engineering efforts", there's the benchmark & dataset track.
> > > >
> > > > > demonstrating convincingly that it is effective for EAI on several tasks
> > > >
> > > > Sure. But was a new method used or were existing methods applied to a new dataset? IMHO the former is suitable for a NeurIPS paper, the latter isn't.
> > > >
> > > > But okay, this is not a hill I'm going to die on, so I'll update my score to also recommend acceptance and the AC can decide if this paper is in the right NeurIPS track.

---

> > > > ### Author Response · Authors · 2022-08-09
> > > > **Reviewer AqAB Rebuttal Response**
> > > >
> > > > We thank the reviewer for their detailed response and positive feedback.
> > > >
> > > > Based on your comments, we are looking into interpretability studies to better understand what is being learnt from our trained agents in HM3D, and running more fine-tuning experiments in HM3D under different ablations. Unfortunately, we do not have enough time to draw meaningful conclusions from these experiments yet, but these questions are being actively pursued by us.
> > > >
> > > > > For the final version of the paper, it would be nice to have the navigation + interaction simulation speed comparison with Habitat-2.0 / iGibson.
> > > >
> > > > Thank you for the thoughtful suggestion. We will do our best to provide a meaningful comparison with the additional platforms in the final paper.

---

> > ### Author Response · Authors · 2022-08-02
> > **Reviewer AqAB Response - Part 3**
> >
> > > W3 and Q3) How do rendering speeds compare to other frameworks like
> > > AI2Thor, iGibson, Gibson, Habitat, Habitat-2.0, etc?
> >
> > Before moving to other comparisons, we should first say: ProcTHOR is
> > built within AI2-THOR and is identical in speed to AI2-THOR. The only
> > complication here is that ProcTHOR houses can vary significantly in
> > size and, as shown in Table 1, larger houses generally result in lower
> > FPS. The iTHOR scenes from AI2-THOR are all one-room houses and are
> > approximately equivalent to the "Small" houses from Table 1.
> >
> > Regarding other comparisons, this is a great question and is
> > surprisingly challenging to answer for several reasons:
> >
> > 1. Different simulators support different agents, each with their
> > own action spaces and capabilities, with little standardization
> > across simulators. AI2-THOR, and thus ProcTHOR as well, supports
> > three different agent types: "high-level", "locobot", and "arm".
> > The  "arm" agent is often slower to simulate than the navigation-only
> > "locobot" agent as it is more complex to physically model a 6 DoF arm
> > as it interacts with objects. This is made even more complex when
> > noting that random action sampling, the simplest policy with which
> > to benchmark, is a poor profiling strategy as some actions are only
> > computationally expensive in rare, but important, settings;
> > for instance, computing arm movements is most expensive when the
> > arm is interacting with many objects, these interactions are rare
> > when randomly sampling but we'd expect them to dominate when using
> > a well-trained agent.
> >
> > 2. Some simulators are relatively slow when run on a single process
> > but can be easily parallelized with many processes running on a
> > single GPU, e.g. AI2-THOR. Thus single-process simulation speeds
> > may be highly deceptive as they do not capture the ease of scalability.
> >
> > 3. When training agents via reinforcement learning, there are a large
> > number of factors that bottleneck training speed and so the value
> > of raw simulator speed is substantially reduced. These factors include:
> >    - Model forward pass when computing agent rollouts.
> >    - Model backward pass when computing gradients for RL losses.
> >    - Environment resets - for many simulators (e.g. ProcTHOR, Habitat)
> >    it is orders of magnitude more expensive to change a scene than
> >    it is to take a single agent step. This can be extremely problematic
> >    when using synchronous RL algorithms as all simulators will need to
> >    wait for a single simulator when that simulator is resetting. When
> >    training this means that, in practice, important "tricks" are employed
> >    to ensure that scene changes are infrequent or synchronized, without
> >    these tricks, performance may be dramatically lower.
> >
> > To attempt to control for the above factors, we set up two profiling
> > experiments, one in Habitat HM3D and one using ProcTHOR-10K, where we:
> >
> > - Use a 2-GPU machine (GeForce RTX 2080 GPUs) where GPU-0 is reserved for
> > the agent's actor-critic policy network and GPU-1 is reserved for
> > simulator instances.
> >
> > - Train agents for the ObjectNav task (using the same LoCoBot agent with
> > the same action space).
> >
> > - For both agents, use the same actor-critic policy network, the same
> > referenced in the paper.
> >
> > - Remove the "End" action so that agents always take the maximum 500
> > steps, this minimizes dependence on the learned policy.
> >
> > - Use a rollout length of 128 with the same set of training
> > hyperparameters across both models.
> >
> > - Use a total of 28 parallel simulator processes, this approximately
> > saturates GPU-1 memory. We found that Habitat instances used
> > slightly less GPU memory than ProcTHOR instances and so we could
> > likely increase the number instances for Habitat slightly, but we
> > kept these equal for more direct comparison.
> >
> > - Use a scene update "trick" which forces all simulators to advance to
> > the next scene in a synchronous fashion after every 10 rollouts (e.g.
> > after every 10 x 128 x 28 = 35,840 total steps across all simulators).
> >
> > We ran the above profiling experiments for ~1M steps and we found that
> > training with Habitat resulted in FPS ranging between 119.7-264.3
> > (230.5 average) and training with ProcTHOR resulted in FPS ranging
> > between 145.5-179.4 (167.7 average). Training in ProcTHOR is thus
> > slower than in Habitat but, for the above set up, this difference
> > is around 1.4x rather than what the difference in single process
> > rendering speed would suggest. While we did not have the time to
> > profile Gibson, iGibson, or Habitat-2.0 in this rebuttal period,
> > these simulators are generally stated to have single-process
> > rendering speeds between AI2-THOR and Habitat and so we expect
> > their FPS numbers between the two above ranges.

---

> > ### Author Response · Authors · 2022-08-02
> > **Reviewer AqAB Response - Part 4**
> >
> > > W4 and Q4) What is being transferred when the visual appearance is
> > > significantly different (like HM3D-Semantic ObjectNav)?
> >
> > This is a very interesting question. We conjecture that large-scale
> > pre-training enables the learning of useful navigation primitives
> > that rely less on scene memorization due to the diversity and scale
> > of the pre-training dataset. When evaluating 0-shot on visually
> > in-domain data (iTHOR, RoboTHOR and ArchitecTHOR), agents perform
> > extremely well, often outperforming past SOTA models that relied
> > on the training data from those benchmarks. HM3D on the other hand
> > can be considered out of domain from a visual standpoint.
> > This likely leads to less impressive 0-shot performance. However,
> > training for a few million steps tunes the model towards the visual
> > attributes of HM3D’s observations and when this is combined with
> > the navigation primitives learnt during training, leads to improved
> > results.
> >
> > Furthermore, based on empirical observations, pre-training on ProcTHOR
> > seems to avoid overfitting during fine-tuning, possibly due to the
> > transfer of navigation abilities from one simulator to another. Here
> > are the training curves on HM3D from a model pre-trained with ProcTHOR
> > compared to one trained from scratch:
> > [https://procthor-rebuttal.netlify.app/hm3d-curves](https://procthor-rebuttal.netlify.app/hm3d-curves).
> >
> > > W5 and Q5) Do the scaling ablations hold true when models are
> > > finetuned? Does the lack of consistent scaling for HM3D-Semantic
> > > ObjectNav reflect poorly on the ability to use ProcThor to benefit
> > > real-world robotics?
> >
> > Table 3 presents ablation results in a 0-shot setting in order to
> > avoid having to fine-tune 16 different models, which would be
> > computationally very expensive. However, the reviewer does ask a
> > valid research question, and hence we present numbers for 10 and
> > 10k ProcTHOR pre-trained models when fine-tuned on RoboTHOR for
> > the task of ObjectNav. As seen, jumping from 10 to 10k provides
> > a huge improvement not just for 0-shot but also for fine-tuning.
> >
> > | Number of Training Houses |RoboTHOR Fine-Tuned Success Rate | RoboTHOR Fine-Tuned SPL |
> > | ----------- | ----------- |----------- |
> > | 10 | 37.2% | 0.303 |
> > |10,000 | 56% | 0.441|
> >
> > As mentioned in W4 above, 0-shot numbers for HM3D aren't as impressive
> > as the other benchmarks, likely due to the visually out of domain
> > nature of HM3D compared to ProcTHOR. Note that ProcTHOR pre-training
> > is still very beneficial, but requires a little bit of fine-tuning
> > to shift the model towards HM3D visually. In light of this, we do
> > not recommend reading too much into the 0-shot HM3D improvements
> > from 10 to 10k houses; they are all fairly low and differences here
> > are less meaningful.

---

### Official Review · Reviewer_L9s9 · 2022-07-12

**Rating:** 4
**Confidence:** 4
**Soundness:** 3 good
**Presentation:** 4 excellent
**Contribution:** 2 fair

**Summary:**

This work presents ProcTHOR, a procedural generation schema to generate synthetic house scenes subject to user-specified constraints. ProcTHOR has been applied to generate a set of 10,000 diverse house environment (fully equipped with sampled objects, materials, and other physics/rendering attributes). Training on these procedurally generated environments achieves state-of-the-art results over a range of embodied-AI tasks that solely rely on RGB images.

**Questions:**

I would like to see **W1**, **W2**, and **W3** discussed in the rebuttal phase

**Limitations:**

I find the discussion satisfactory

**Strengths And Weaknesses:**

Strengths
=========

**S1** Embodied-AI is a very active area of research in the robot learning and computer vision communities. The tasks addressed in this paper are directly based on a number of benchmarks proposed over the last 5 years. I thus expect this work to be relevant to several members at Neurips.

**S2** This paper is extremely well-written. It is easy to follow the core contributions of this work, and the range of experiments (both in terms of environments and tasks) is somewhat broad. I also skimmed through some of the technical details about the dataset and I appreciated all of the thought and work that went into preparing the manuscript and supplemntary materials. Overall, this is thoroughly well-executed work.

---

Weaknesses
==========

While the work itself is well-executed, I have concerns over the technical aspects of this paper, which need further discussion.

**W1** *Core contribution a dataset?* The paper, in its current form, comes off as a dataset contribution: the core takeaway for me was that it is possible to procedurally generate as many scenes as desired; and that the combination of the procedural generation scheme and the scale it achieves results in state-of-the-art performance over a number of embodied-AI tasks. While this is a demonstration that scale (and to an extent domain randomization) results in good performance, the key contribution enabling this is the ProcTHOR generation technique, which is more of a dataset (generator). I feel that the "datasets and benchmarks" track at Neurips would have been a much more suitable fit for this paper. I would like to see this aspect discussed further in the author-response phase. Particularly because, the neurips guidelines (here)[https://neurips.cc/Conferences/2022/CallForDatasetsBenchmarks] and (here)[https://neurips.cc/Conferences/2022/ReviewerGuidelines] focus on "algorithmic advances, analysis, and applications"; and I have a hard time reconciling with either of these categories in terms of technical contributions of this submission.

**W2** *Focus on RGB-only agents*: The current submissions only evaluate RGB-only agents across environments. While this is arguably a harder task than the scenario where RGB-D data is available, several researchers interested in using the ProcTHOR environment are also likely interested in leveraging RGB-D sensors. It is therefore important to analyze performance on various other sensor configurations, and evaluate the gains ascribed by ProcTHOR when varying sensing modalities.

**W3** *Procedural generation*: Building on **W1**, if the procedural generation scheme itself were to be considered a core contribution, it would have helped to have evaluation with respect to other competing procedural generation schemes (such as those cited in Sec. 2), or with respect to the various design choices within the ProcTHOR environment itself.

---

(Another aspect that could be relevant for future exploration is to optimize aspects of the procedural generator itself, to maximize performance on a set of downstream tasks. A somewhat recent example of this is observed in "Meta-Sim: Learning to Generate Synthetic Datasets").

---

> ### Author Response · Authors · 2022-08-02
> **Reviewer L9s9 Response - Part 1**
>
> Thank you for the constructive and insightful feedback. We appreciate
> the supportive comments about the paper being “extremely well-written”
> and the work being “thoroughly well-executed”. We address your concerns
> below.
>
> > W1) Why not submit to the datasets track?
>
> We have addressed this concern in the Overall Response above. Please
> refer to that response.
>
> > W2) Focus on RGB-only agents
>
> ProcTHOR supports all sensors supported by AI2-THOR, which includes RGB,
> depth, segmentation masks, etc. Our primary motivation to evaluate
> RGB-only agents were to choose the hardest sensor configuration (as you
> mentioned) as well as choose the sensors that were most easily available
> and reliable in the real-world. While depth sensors are available, they
> work at lower resolutions than RGB, still tend to be fairly noisy and
> often have missing depth information across several pixels and regions
> in the image.
>
> We also note that our presented RGB-only results outperform past works,
> some of which also include more sensors such as depth.
>
> In response to your request, we provide experimental results for
> ArmPointNav employing the RGB-D sensor. Due to time limitations, we
> have run the RGBD training for 44M frames (vs 100M in the original
> paper). The experiment is still running; we will update the numbers
> with the latest results. The results thus far align with
> the findings of [1, 2] that RGB-D doesn’t provide significant
> improvements over using RGB only using present-day models and that
> future work should investigate new architectures that can use RGB
> and Depth information more effectively.
>
> |    Training on  | PuSR |SR |
> | ----------- | ----------- |----------- |
> | ProcTHOR-RGB @ 100M      | 74.8 |37.9|
> | ProcTHOR-RGBD @ 44M   |68.3 |33.1||
>
> Edit (Aug 9): Training to 100M steps with RGBD does not improve numbers meaningfully. As noted previously, this aligns with findings from [1, 2].
>
> [1] Ehsani, K., Han, W., Herrasti, A., VanderBilt, E., Weihs, L., Kolve, E., ... & Mottaghi, R. (2021). Manipulathor: A framework for visual object manipulation. In Proceedings of the IEEE/CVF conference on computer vision and pattern recognition (pp. 4497-4506).
>
> [2] Deitke, M., Han, W., Herrasti, A., Kembhavi, A., Kolve, E., Mottaghi, R., ... & Farhadi, A. (2020). Robothor: An open simulation-to-real embodied ai platform. In Proceedings of the IEEE/CVF conference on computer vision and pattern recognition (pp. 3164-3174).

---

> > ### Author Response · Authors · 2022-08-02
> > **Reviewer L9s9 Response - Part 2**
> >
> > > W3) Procedural generation evaluation: it would have helped to have evaluation with respect to other competing
> > procedural generation schemes (such as those cited in Sec. 2), or with
> > respect to the various design choices within the ProcTHOR environment
> > itself.
> >
> > Thank you for the suggestion. We have added a detailed comparison of our
> > work to others in the literature in Appendix B.12, including discussions
> > of potential tradeoffs with alternative approaches. Based on the results
> > in this work, we conjecture that there are many possible procedural
> > generation schemes that can be used to effectively train agents. The
> > goal of our work was to build an initial algorithm that generates diverse
> > and semantically plausible houses using the best existing approaches or
> > building on existing works in areas that were insufficient for our use
> > case. At every step in building ProcTHOR, we performed local studies of
> > various algorithms and chose the approach that, qualitatively, resulted
> > in high-quality generations.
> >
> > Our floorplan generation algorithm is based on [1], which provides a way
> > to procedurally generate diverse and plausible floorplans without any
> > external data. We chose this approach because it only requires a room
> > spec and an interior boundary, and doesn’t rely on an external database
> > of floorplans to synthesize one. Thus, it is trivial to scale to include
> > new room types (e.g., garages, balconies, stairways) and generate any
> > type of home (e.g., from studio apartments to massive multi-family homes)
> > just by modifying the room specs. [2, 3, 4] train a network to generate
> > floorplans, but they do not support inputting any preferences about the
> > number of rooms or the types of rooms in the scene. [5] supports passing
> > in constraints, but it cannot generalize to new room types not seen
> > during training, or to massive multi-family homes.
> >
> > Most work on object placement [6, 7, 8, 9] leverages priors about where
> > objects are placed in large 3D scene datasets, such as 3D-Front or SUNCG.
> > The works assume a fixed object database while training the priors and
> > generating novel scenes. Therefore, we cannot easily adapt such approaches
> > to our work as ProcTHOR’s object database is completely different and
> > our database does not have massive amounts of 3D scenes with example
> > object placements.
> >
> > Given the huge engineering effort and computational costs in designing
> > our procedural generation system, generating the ProcTHOR-10k dataset,
> > and training a large collection of embodied AI models; a comprehensive
> > quantitative ablative study of different generation algorithms is beyond
> > the scope of this work.
> >
> > [1] Lopes, R., Tutenel, T., Smelik, R. M., De Kraker, K. J., & Bidarra, R. (2010, November). A constrained growth method for procedural floor plan generation. In Proc. 11th Int. Conf. Intell. Games Simul (pp. 13-20). Citeseer.
> >
> > [2] Nauata, N., Chang, K. H., Cheng, C. Y., Mori, G., & Furukawa, Y. (2020, August). House-gan: Relational generative adversarial networks for graph-constrained house layout generation. In European Conference on Computer Vision (pp. 162-177). Springer, Cham.
> >
> > [3] Nauata, N., Hosseini, S., Chang, K. H., Chu, H., Cheng, C. Y., & Furukawa, Y. (2021). House-gan++: Generative adversarial layout refinement network towards intelligent computational agent for professional architects. In Proceedings of the IEEE/CVF Conference on Computer Vision and Pattern Recognition (pp. 13632-13641).
> >
> > [4] Wu, W., Fu, X. M., Tang, R., Wang, Y., Qi, Y. H., & Liu, L. (2019). Data-driven interior plan generation for residential buildings. ACM Transactions on Graphics (TOG), 38(6), 1-12.
> >
> > [5] Hu, R., Huang, Z., Tang, Y., Van Kaick, O., Zhang, H., & Huang, H. (2020). Graph2plan: Learning floorplan generation from layout graphs. ACM Transactions on Graphics (TOG), 39(4), 118-1.
> >
> > [6] Zhang, S. K., Xie, W. Y., & Zhang, S. H. (2021). Geometry-based layout generation with hyper-relations among objects. Graphical Models, 116, 101104.
> >
> > [7] Wang, K., Lin, Y. A., Weissmann, B., Savva, M., Chang, A. X., & Ritchie, D. (2019). Planit: Planning and instantiating indoor scenes with relation graph and spatial prior networks. ACM Transactions on Graphics (TOG), 38(4), 1-15.
> >
> > [8] Wang, X., Yeshwanth, C., & Nießner, M. (2021, December). Sceneformer: Indoor scene generation with transformers. In 2021 International Conference on 3D Vision (3DV) (pp. 106-115). IEEE.
> >
> > [9] Paschalidou, D., Kar, A., Shugrina, M., Kreis, K., Geiger, A., & Fidler, S. (2021). Atiss: Autoregressive transformers for indoor scene synthesis. Advances in Neural Information Processing Systems, 34, 12013-12026.
> >
> > > Another aspect that could be relevant for future exploration is to optimize aspects of the procedural generator itself, to maximize performance on a set of downstream tasks. A somewhat recent example of this is observed in "Meta-Sim: Learning to Generate Synthetic Datasets".
> >
> > Thank you for the suggestion! We have incorporated this work into our future work discussion.

---

### Author Response · Authors · 2022-08-02
**Overall Response - Main track vs Datasets track**

A common question that was posed by two reviewers, L9s9 and QxMz, was why
we chose to submit our paper to the main track as opposed to the dataset
and benchmark track.

The decision to submit to the main conference track was taken by the
authors of the paper after a fair bit of deliberation. This paper has
two key contributions: (1) A new framework for procedural generation
with an accompanying artist-designed test set (2) The first large-scale
demonstration of pre-training for Embodied AI to cover 6 benchmarks
across multiple tasks and simulators. The first contribution is indeed
central to the paper and the advantages and usage of ProcTHOR will
likely extend far beyond the 6 benchmark results presented in this
paper. However, as mentioned by all reviewers, the results we have
obtained are very strong and broad, including 0-shot performance on
tasks that even outperforms previous SOTA results. While large scale
studies spanning multiple datasets and tasks are seen more often in
computer vision, it is quite rare to see multiple task and multiple
simulator studies with SOTA results in the domain of Embodied AI.
Another point we would like to note is that our 0-shot results are
surprising and unexpectedly strong, and notable for the Embodied AI
community, where one expects that models trained on procedurally
generated data will simply learn biases within the generation process
and have a tough time generalizing 0-shot to human curated downstream
tasks. The authors felt that given these strong results and unexpected
findings, we should submit this work to the main conference.

---

> ### Comment · Reviewer_QxMz · 2022-08-07
> **Hm**
>
> As mentioned in my response to the R#AqAB below (https://openreview.net/forum?id=4-bV1bi74M&noteId=Uq2dKEhlCyi), I still hold that the main contribution of this work is a dataset and engineering efforts and as such, the paper should go into the benchmarks & dataset track.
>
> But the paper and work itself is of very high quality and should definitely be accepted **somewhere at NeurIPS**, so I'll change my review to acceptance and let the AC comment on the track.

---

> > ### Comment · Area_Chair_AgNm · 2022-08-09
> > **Main track submission is ok**
> >
> >
> > Thanks folks for the robust discussion.
> >
> > Regarding the question of main track vs. the datasets and benchmarks track. I refer to the datasets and benchmarks track [call for papers](https://nips.cc/Conferences/2022/CallForDatasetsBenchmarks) which says "It is also still possible to submit datasets and benchmarks to the main conference (under the usual review process)". Therefore it is the authors' choice which track to submit to and this paper will be considered for the main track.
> >
> > Best,
> > AC

---

### Meta-Review · Area_Chair_AgNm · 2022-08-26

**Recommendation:** Accept
**Confidence:** Certain

**Metareview:**

*Summary*

The paper presents ProcThor, a framework to generate interactive 3D environments from an underlying distribution of room and object layouts. In the current work, 10000 3D environments of varying sizes, # rooms, and object distributions are sampled and enable simulation for object search and manipulation tasks. The experiments demonstrate that training policies on the synthetically generated environments and the finetuning it on other datasets like RoboThor, AI2Thor, and Habitat-Matterport3D lead to state-of-the-art performances. Furthermore, the ablation experiments reveal that the transfer task performance continues to improve as more and more virtual 3D environments are sampled for training. Finally, 'zero-shot' experiments (without finetuning) are included with surprisingly good results.

*Reviews*

The reviewers' ratings are 4 (borderline reject), 8 (strong accept) and 9 (very strong accept).

Reviewer L9s9 (borderline reject) did not respond to the rebuttal, but their main concerns were:
- (W1) the paper should be submitted to the datasets track
- (W2) the paper should include experiments with RGB-D sensors, and
- (W3) the paper should include experiments with other procedural generation schemes.

*Decision*

I have already discussed and dismissed concern W1 in my comments below. Regarding W2 and W3, these are asks for more experiments. The authors provided reasonable justifications and responses to these concerns. Further, I'm persuaded by the other reviewers that the paper already represents a 'massive feat of engineering' and note that the experiments already cover both zero-shot and finetuning settings on multiple leaderboards. Therefore I am putting less weight on L9s9's low rating.

AqAB and QxMz are united that this paper and the associated code release will have a significant impact on the Embodied AI community and I agree. The paper is also acknowledged as being extremely well-written, with very thorough experiments, and I feel it could be considered for an outstanding paper award.



**Award:**

Yes

---

### Decision · Program_Chairs · 2022-09-14

Accept